# Combination of Arsenic Trioxide and Valproic Acid Efficiently Inhibits Growth of Lung Cancer Cells via G2/M-Phase Arrest and Apoptotic Cell Death

**DOI:** 10.3390/ijms21072649

**Published:** 2020-04-10

**Authors:** Hyun Kyung Park, Bo Ram Han, Woo Hyun Park

**Affiliations:** Department of Physiology, Medical School, Research Institute for Endocrine Sciences, Chonbuk National University, 20 Geonji-ro, Deokjin-gu, Jeonju, Jeollabuk 54907, Korea; shenaringo27@naver.com (H.K.P.); ramram@jbnu.ac.kr (B.R.H.)

**Keywords:** arsenic trioxide, valproic acid, lung cancer cells, cell cycle arrest, apoptosis

## Abstract

Arsenic trioxide (ATO; As_2_O_3_) has anti-cancer effects in various solid tumors as well as hematological malignancy. Valproic acid (VPA), which is known to be a histone deacetylase inhibitor, has also anti-cancer properties in several cancer cells including lung cancer cells. Combined treatment of ATO and VPA (ATO/VPA) could synergistically enhance anti-cancer effects and reduce ATO toxicity ATO. In this study, the combined anti-cancer effects of ATO and VPA (ATO/VPA) was investigated in NCI-H460 and NCI-H1299 lung cancer cells in vitro and in vivo. A combination of 3 μM ATO and 3 mM VPA (ATO/VPA) strongly inhibited the growths of both lung cancer cell types. DNA flow cytometry indicated that ATO/VPA significantly induced G2/M-phase arrest in both cell lines. In addition, ATO/VPA strongly increased the percentages of sub-G1 cells and annexin V-FITC positive cells in both cells. However, lactate dehydrogenase (LDH) release from cells was not increased in ATO/VPA-treated cells. In addition, ATO/VPA increased apoptosis in both cell types, accompanied by loss of mitochondrial membrane potential (MMP, ∆Ψm), activation of caspases, and cleavage of anti-poly ADP ribose polymerase-1. Moreover, a pan-caspase inhibitor, Z-VAD, significantly reduced apoptotic cell death induced by ATO/VPA. In the xenograft model, ATO/VPA synergistically inhibited growth of NCI-H460-derived xenograft tumors. In conclusion, the combination of ATO/VPA effectively inhibited the growth of lung cancer cells through G2/M-phase arrest and apoptotic cell death, and had a synergistic antitumor effect in vivo.

## 1. Introduction

Lung cancer, the leading cause of cancer death worldwide among both men and women older than 25, accounts for one quarter of cancer deaths [1,2]. Lung cancer can be divided into two categories: non-small cell lung cancer (NSCLC) and small cell lung cancer (SCLC). NSCLC is diagnosed in 85% of patients with lung cancer and is divided into three major histologic subtypes: adenocarcinoma, squamous cell carcinoma, and large cell carcinoma [3]. Various therapeutic strategies are currently under consideration because the use of cytotoxic drugs is limited by drug toxicity and tumor resistance [4]. An improved understanding of molecular mechanisms underlying anti-cancer drug actions allows new insights into the treatment of lung cancer and the development of innovative agents that target specific intracellular pathways.

Inorganic arsenic has long been utilized as medicine to treat several severe diseases in China and ancient Greece, and especially arsenic trioxide (ATO; As_2_O_3_) has been identified as an effective therapeutic agent in certain leukemia patients without severe bone marrow suppression [5,6]. Further, the therapeutic effect of ATO may be functional against solid tumors since its mechanisms of action are mainly induction of apoptosis and cell cycle arrest [7]. In fact, ATO shows anti-cancer effects in renal [8], head and neck [9], prostate [10], hepatoma [11], cervical [12], and gastric cancer [13] via affecting many biological functions including cell proliferation, differentiation, angiogenesis, and apoptosis. In addition, ATO also inhibits the growth of lung cancer cells via apoptosis [14]. ATO acts as a mitochondrial toxin and can induce the loss of mitochondrial transmembrane potential (MMP, ∆Ψm) [7] and, as such, increases the production of reactive oxygen species, subsequently causing apoptosis in target cells [15,16,17].

Combination therapy is frequently used in clinical studies to improve the beneficial effects and reduce the toxicity of anti-cancer agents [18,19]. Higher dose of ATO is toxic and induces side effects in hepatic, cardiovascular, and nervous systems [20,21]. Therefore, the combination treatment of ATO with other agents have been reported to inhibit cell growth and induce apoptosis in various cancer cells including lung cancer [22,23,24]. Epigenetic mistakes contribute to the initiation and progression of cancer and to response to therapy of cancer [25]. Histone deacetylase (HDAC) inhibitors induce antitumor activity by withdrawing epigenetic errors [26,27,28]. In particular, valproic acid (VPA), which was originally used to treat bipolar disorder and epilepsy, is known to be an inhibitor of HDACs displaying applicable pharmacokinetic properties, and yielding only moderate harmfulness that is acceptable in the perspective of an anticancer treatment [29,30,31,32]. By modulating a broad range of activities, including cell proliferation, apoptosis, differentiation, and metastasis, VPA has anti-cancer properties in several cancers, including lung cancer cells [31,33,34,35]. 

Thus, combination of ATO and VPA (ATO/VPA) could act synergistically to enhance anti-cancer effects and reduce the toxicity of ATO. However, there is not report about such combined effects in lung cancer. In the present study, the combined effects of ATO/VPA was assessed in NCI-H460 and NCI-H1299 lung cancer cells in relation to cell growth, cell death (i.e., necrosis and apoptosis), and cell cycle arrest.

## 2. Results

### 2.1. Effects of ATO or VPA on Cell Growth and Necrotic Cell Death in Lung Cancer Cells

The effect of ATO or VPA on the growths of NCI-H460 and NCI-H1299 lung cancer cell types was observed using MTT assays. ATO dose-dependently inhibited the growth of NCI-H460 cells, with IC_50_ values of >10 and 6 μM after 24 and 72 h of incubation, respectively (Figure 1A). Similarly, IC_50_ values for VPA were 8 and 4 mM in NCI-H460 cells at 24 and 72 h, respectively (Figure 1A). Using NCI-H1299 cells, IC_50_ values of 4 μM and 7 mM at 72 h were obtained for ATO and VPA, respectively. (Figure 1B). Neither drug caused growth inhibition of NCI-H1299 cells at 24 h (Figure 1B). Once cells are impaired by stress, injuries, or intercellular signals, lactate dehydrogenase (LDH) is rapidly released from the cell membrane, which is relatively a biomarker for necrotic cell death rather than apoptotic cell death. Treatment with 3–10 μM ATO did not significantly affect LDH release in NCI-H460 and NCI-H1299 cells at 24 h but 10 μM ATO seemed to increase LDH release in both lung cancer cells at 72 h (Figure 1C,D). LDH release in NCI-H460 cells treated with 10 mM VPA increased at 24 and 72 h (Figure 1E). In contrast, VPA did not affect LDH release in NCI-H1299 at 24 and 72 h (Figure 1F). The current results indicated that relatively longer and higher exposure of ATO or VPA induced necrotic cell death in NCI-H460 cells. 

### 2.2. Effects of ATO and VPA Alone and in Combination on Cell Morphology and Cell Cycle Distributions

Morphologic changes were visualized by inverted microscopy for NCI-H460 and NCI-H1299 cells that were treated with 3 μM ATO and 3 mM VPA alone and in combination after 72 h incubation. The concentrations of 3 μM ATO and 3 mM VPA and the incubation time of 72 h were considered as suitable doses and time to distinguish the differences of dead and live cells. Compared with groups treated with single drugs only, a decrease in cell number was observed after treatment with ATO/VPA (Figure 2A). Growth inhibition can be explained by an arrest during cell cycle progression, therefore cell cycle distributions were also analyzed at 72 h (Figure 2B). While 3 μM ATO induced G2/M-phase arrest of the cell cycle in both NCI-H460 and NCI-H1299 cells, 3 mM VPA induced G1-phase arrest in NCI-H1299 cells (Figure 2B,C). In addition, ATO/VPA significantly increased the proportions of G2/M-phase cells in both NCI-H460 and NCI-H1299 cells (Figure 2B,C).

### 2.3. Effects of ATO and VPA Alone and in Combination on Cell Death, LDH Release, and Apoptosis

ATO/VPA (3 μM ATO and 3 mM VPA) significantly increased the percentages of sub-G1 cells in both NCI-H460 and NCI-H1299 cells (Figure 2D). LDH release was not increased in NCI-H460 and NCI-H1299 cells after treatment with ATO/VPA (Figure 2E). Whether ATO and VPA induces apoptotic cell death in cells was evaluated using annexin V-FITC/PI staining cells. Treatment with 3 mM VPA significantly increased the number of annexin V-staining cells in NCI-H460 cells whereas 3 μM ATO augmented the number in NCI-H1299 cells (Figure 3A,B). The percentages of annexin V-staining cells in NCI-H460 and NCI-H1299 cells treated with ATO/VPA were synergistically increased, compared with cells treated with single drugs alone (Figure 3A,B). In addition, changes in apoptosis-related proteins were detected with Western blotting. The intact form of PARP was clearly reduced in both NCI-H460 and NCI-H1299 cells after treatment with ATO/VPA, and the cleavage form of PARP was strongly induced in these cells. The cleavage forms of caspase-3, caspase-8, and caspase-9 also clearly increased in ATO/VPA-treated cells, compared with cells treated with single drugs alone (Figure 3C).

### 2.4. Effects of ATO and VPA Alone and in Combination on MMP (∆Ψm)

As apoptosis is closely related to the collapse of MMP (∆Ψm), loss of MMP (∆Ψm) in cells was evaluated using JC-1 and Rhodamine 123 dyes. When NCI-H1299 cells were treated with ATO and VPA alone and in combination, JC-1 green fluorescence in ATO/VPA-treated cells was more intense than fluorescence in cells treated with single drugs alone (Figure 4A). VPA significantly increased the loss of MMP (∆Ψm) in NCI-H460 cells (Figure 4B). In ATO/VPA-treated cells, MMP (∆Ψm) loss was significantly increased in both cell lines (Figure 4B,C).

### 2.5. Effect of Z-VAD on Cell Death in Lung Cancer Cells

Whether Z-VAD (a pan-caspase inhibitor) affects cell death in NCI-H460 and NCI-H1299 cells after treatment with ATO/VPA was investigated. Both cell lines were pre-incubated with or without 15 μM Z-VAD for 1 h before treatment with ATO/VPA. Z-VAD significantly reduced percentages of sub-G1 cells in both ATO/VPA-treated NCI-H460 and NCI-H1299 cells, compared with cells treated with ATO or VPA alone (Figure 5A). However, The Z-VAD did not significantly affect LDH release in NCI-H460 and NCI-H1299 cells after treatment with ATO/VPA (Figure 5B). Similar to the reduction of sub-G1 cells by Z-VAD, this inhibitor decreased the percentages of annexin V-staining cells in ATO/VPA-treated NCI-H460 and NCI-H1299 cells (Figure 5C).

### 2.6. Anti-Tumor Effect of ATO and VPA Alone and in Combination on NCI-H460 Xenograft Nude Mice

To investigate whether ATO/VPA has anti-tumor effects in vivo, lung cancer xenografts using BALB/c nude mice were examined. At the end of the study, mice were sacrificed, and tumors were removed (Figure 6A). Body weight was not affected by drug administration (Figure 6B). Treatment with single drugs alone did not significantly decrease tumor volume during the experimental period (Figure 6C,D). However, the ATO/VPA-treated group showed significant inhibition of tumor growth (Figure 6C,D).

## 3. Discussion

In the current study, ATO and VPA inhibited growth of NCI-H460 and NCI-H1299 lung cancer cells in dose- and time-dependent manners. A combination of 3 μM ATO and 3 mM VPA (ATO/VPA) strongly inhibited the growths of both lung cancer cell types. DNA flow cytometry indicated that VPA induced G1-phase arrest in NCI-H1299 cells and ATO induced G2/M-phase arrest in both NCI-H460 and NCI-H1299 cells. In addition, the combination of ATO/VPA also induced G2/M-phase arrest in both cell lines. ATO is known to induce G1 or G2/M-phase arrest depending on cell type [14,36,37]. Similarly, VPA can induce G1 or G2/M-phase arrest in glioblastoma, cervical cancer, and gastric cancer cells [31,38,39]. Molecular mechanisms of cell cycle arrest by ATO or VPA thus vary depending on cell type. Cell cycle arrest, especially in G2/M phase in ATO/VPA-treated lung cancer cells, is likely to be an underlying mechanism for suppression of cell growth.

The combination of ATO/VPA significantly increased the percentages of sub-G1 cells and the number of annexin V-FITC positive cells in both NCI-H460 and NCI-H1299 cells. However, LDH release was not significantly increased by ATO/VPA. Thus, ATO/VPA appeared to synergistically induce apoptotic cell death in lung cancer cells, but this combination relatively did not trigger necrotic cell death. Generally, apoptosis involves two signaling pathways: cell death receptor (extrinsic) and mitochondrial (intrinsic) pathways [40]. Cell death receptor in response to TRAIL, TNFα, and Fas activates caspase-8. Activated caspase-8 cleaves cytosolic BID to generate a truncated product (tBID) that translocates to the mitochondria and decreases MMP (∆Ψm). The key step in the mitochondrial pathway is the efflux of cytochrome c from mitochondria to cytosol. In the cytosol, cytochrome c forms a complex called apoptosome with Apaf-1 and caspase-9 that activates other caspases including caspase-3 and caspase-7 [41]. Consequently, crosstalk between apoptotic pathways is facilitated through caspase-3. Caspase-3 is an executioner caspase that can systematically dismantle cells by cleaving key proteins such as PARP when activated. When lung cancer cells were treated with ATO/VPA, cleaved forms of caspase-8 and caspase-9 were increased, implying that caspase-8 linked with the cell death receptor (extrinsic) pathway was activated in these cells, and caspase-9 related to the mitochondrial (intrinsic) pathway was also activated. In addition, cleaved form of executive caspase-3, which means the activation of caspase-3, was clearly detected in lung cancer cells treated with ATO/VPA. Thus, apoptotic cell death induced by ATO/VPA apparently involves both intrinsic and extrinsic pathways.

Apoptosis is closely related to the failure of MMP (∆Ψm) [42]. The combination of ATO/VPA synergistically or additively triggered loss of MMP (∆Ψm) compared with treatment with single drugs. Apoptotic cell death induced by ATO/VPA is thus connected to the collapse of MMP (∆Ψm). Numbers of sub-G1 and annexin V-FITC positive cells were effectively lowered by Z-VAD in ATO/VPA-treated NCI-H460 and NCI-H1299 cells. However, The Z-VAD did not influence the level of LDH release in these cells. Therefore, ATO/VPA efficiently triggers apoptotic cell death rather than necrotic cell death.

Previous reports demonstrate that ATO (3–5 mg/Kg) causes decreases in growth of tumors from cervical and liver cancer cells [43,44] and VPA (200–500 mg/kg) also inhibits growth of tumors from prostate, pancreatic, and lung cancer cells [45,46,47]. In the xenograft model tested in this experiment, there is no difference in body weight of nude mice treated with ATO (5 mg/Kg), VPA (400 mg/Kg), or combined ATO/VPA. Therefore, the used doses of ATO and VPA were likely to be toxicologically tolerable in BALB/c nude mice. ATO significantly inhibited tumor growth cells, and VPA alone had a similar but weaker effect in the xenograft mouse. The combination of ATO/VPA showed significant and synergic effects on the inhibition of tumor growth in xenografts from H460 lung cancer cell line. In vitro synergistic effects on cell growth inhibition by ATO/VPA are thus reproduced with in vivo experiments.

In conclusion, the combination of ATO/VPA effectively inhibited the growth of NCI-H460 and NCI-H1299 lung cancer cells via G2/M-phase arrest of the cell cycle and caspase-dependent apoptosis-related to the loss of MMP (∆Ψm) (Figure 7). ATO/VPA had a synergistic antitumor effect in vivo. Combining ATO and VPA can be a hopeful strategy in clinical therapy for lung cancer. These findings add important information to our understanding of the synergistic effects of ATO/VPA on lung cancer cells.

## 4. Materials and Methods

### 4.1. Cell Culture

Human lung cancer NCI-H460 and NCI-H1299 cells were obtained from the American Type Culture Collection (Manassas, VA, USA). Cells were cultured in RPMI-1640 medium (Hyclone, Logan, UT, USA) containing 1% penicillin–streptomycin (GIBCO BRL, Grand Island, NY, USA) and 10% fetal bovine serum (GIBCO BRL). Cells were grown in 10 cm plastic cell culture dishes (BD Falcon, Franklin Lakes, NJ, USA) and harvested with a trypsin-EDTA solution (GIBCO BRL).

### 4.2. Reagents

ATO and VPA individually were purchased from Sigma-Aldrich Co. (St. Louis, MO, USA). Stock solutions of ATO (10^−1^ M) dissolved in 1.65 M NaOH and of VPA (1 M) dissolved in distilled water were prepared. The pan-caspase inhibitor (Z-VAD-FMK; benzyloxycarbonyl-Val-Ala-Asp-fluoromethylketone) was obtained from R&D Systems Inc. (Minneapolis, MN, USA) and a stock solution (10 mM) in dimethyl sulfoxide (DMSO; Sigma-Aldrich Co.) was prepared.

### 4.3. Cell Growth Inhibition Assay

The effect of ATO or VPA on cell growth inhibition was determined by measuring 3-(4,5-dimethylthiazol-2-yl)-2,5-diphenyltetrazolium bromide (MTT; Sigma-Aldrich Co.) absorbance in living cells, as previously described [48]. In brief, 5 × 10^3^ cells were seeded in 96-well microtiter plates (SPL Life Science, Pocheon, Korea). After treatment with ATO or VPA for the indicated times, 20 µL of MTT solution (2 mg/mL) was added to each well. Plates were incubated for an additional 3 h at 37 °C. Media were removed from plates, and 200 µL DMSO was added to each well to solubilize formazan crystals. Optical density was measured at 570 nm using a microplate reader (Synergy^TM^2; BioTek Instruments Inc., Winooski, VT, USA).

### 4.4. Lactate Dehydrogenase (LDH) Cytotoxicity Assay

The measurement of the amount of LDH released from cells is one of the major methods to assess cell death, especially necrotic cell death. Released lactate dehydrogenase (LDH) activity was measured using an LDH cytotoxicity assay kit (DoGenBio Co., Seoul, Korea). Briefly, 1 × 10^5^ cells in a six-well plate (BD Falcon) were incubated with ATO and VPA alone and in combination with ATO/VPA for 24 and 72 h. After treatment, culture media was collected and centrifuged for 5 min at 1500 rpm. The supernatant was dispensed into a 96-well plate, and LDH assay reagent was added to each well. The plate was incubated at room temperature for 30 min. Absorbance values were measured at 450 nm using a microplate reader (Synergy^TM^2). LDH release was expressed as the percentage of extracellular LDH activity compared with the control group.

### 4.5. Observation of Cell Morphologic Changes

NCI-H460 and NCI-H1299 cells were seeded at a density of 1 × 10^5^ cells in six-well plates. Cells were treated with ATO and VPA alone and in combination for 72 h. Cell population and morphologic changes were determined using inverted microscopy at 40× magnification (Olympus CKX-41, Tokyo, Japan).

### 4.6. Cell Cycle Distribution Analysis

Cell cycle and sub-G1 analysis were determined by propidium iodide (PI, Ex/Em = 488/617 nm; Sigma-Aldrich Co.) staining, as previously described [49]. In brief, 1 × 10^5^ cells in a six-well plate were incubated with the designated doses of ATO and VPA alone and in combination for 72 h. Total cells including floating cells were washed with phosphate-buffered saline (PBS) twice and incubated with PI (10 µg/mL) and RNase (Sigma-Aldrich Co.) at 37 °C for 30 min. Cellular DNA contents were measured using an Accuri C6 flow cytometer (BD Sciences, Franklin Lakes, NJ, USA), and cell cycle distributions were analyzed using BD Accuri^TM^ C6 software (BD Sciences).

### 4.7. Annexin V/PI Staining for Apoptosis Detection

Apoptosis was detected by staining cells with annexin V-fluorescein isothiocyanate (FITC, Miltenyi Biotec, Bergisch Gladbach, Germany; Ex/Em = 488 nm/519 nm) and PI staining, as previously described [48]. Briefly, 1 × 10^5^ cells were incubated in a six-well plate with the indicated doses of ATO and VPA alone and in combination for 72 h with or without Z-VAD. Cells were washed with PBS twice and then suspended in 200 µL of binding buffer (10 mM HEPES pH 7.4, 140 mM NaCl, 2.5 mM CaCl_2_). Annexin V-FITC (2 µL) and PI (1 µg/mL) were then added, and cell suspensions were analyzed with an Accuri C6 flow cytometer (BD Sciences).

### 4.8. Western Blot Analysis

Protein expression levels were evaluated by Western blot analysis, as previously described [28]. In brief, 5 × 10^5^ cells were incubated in 60 mm culture dishes (BD Falcon) with the indicated doses of ATO and VPA alone and in combination for 24 and 72 h. Cells were washed with PBS and then suspended in five volumes of lysis buffer (PRO-PREP^TM^ Protein Extraction Solution; Intron Biotechnology, Seongnam, Korea). Protein concentrations were determined using the Bradford (Bio-Rad, Hercules, CA, USA) method. Supernatant samples containing 30 µg total protein were resolved in 12.5% and 15% SDS-PAGE gels, transferred to Immobilon-P PVDF membranes (Millipore, Bedford, MA, USA) by electroblotting, and probed with anti-poly(ADP-ribose) polymerase (PARP), anti-cleaved PARP, anti-cleaved caspase-3, anti-cleaved caspase-8, anti-cleaved caspase-9 (Cell Signaling Technology Inc., Danvers, MA, USA), and anti-GAPDH antibodies (Santa Cruz Biotechnology, Santa Cruz, CA, USA). Membranes were incubated with horseradish peroxidase-conjugated secondary antibodies (Enzo Life Sciences, Farmingdale, NY, USA). Blots were developed using an ECL kit (DoGen Bio Co., Seoul, Korea).

### 4.9. Measurement of MMP (∆Ψm) Levels

MMP (∆Ψm) levels were measured using JC-1 (Enzo Life Sciences; Ex/Em = 515 nm/529 nm) and rhodamine123 (Sigma-Aldrich Co.; Ex/Em = 485 nm/535 nm) dyes. In brief, 5 × 10^4^ cells in a 12-well culture plate (BD Falcon) were incubated with the indicated concentrations of ATO and VPA alone and in combination for 72 h. Cells were washed twice with PBS and incubated with 10 μg/mL JC-1 or 0.1 μg/mL rhodamine 123 at 37 °C for 30 min. To stain nuclei, cells were incubated with 500 nM 4′, 6′-diamidino-2-phenylindole (DAPI, Life Technologies, Ex/Em = 358 nm/461 nm) at 37 °C for 30 min. After incubation with JC-1 and DAPI, cells were washed twice with PBS, and images were captured by fluorescence microscopy (FLoid^®^ Cell Imaging Station, Life Technologies) at 400× magnification. Green fluorescence indicates lower MMP (∆Ψm), and red fluorescence indicates higher MMP (∆Ψm). The intensity of rhodamine 123 staining was determined by Accuri C6 flow cytometry (BD Sciences). Rhodamine 123 negative cells indicate the loss of MMP (ΔΨ_m_).

### 4.10. Lung Cancer Xenograft Model

Four week-old female BALB/c nude mice were purchased from Nara-Biotec (Seoul, Korea). All experiments were conducted in accordance with a protocol approved by the Institutional Animal Care and Use Committee of Chonbuk National University (CBNU 2016-95. 1 November 2016). NCI-H460 cells (2 × 10^6^) were subcutaneously (s.c.) injected with 100 μL Matrigel (Sigma-Aldrich Co.) and 100 μL PBS into the flanks of mice. After 10 days, PBS 200 µL (control), 5 mg/kg ATO, 400 mg/kg VPA, and combination of ATO/VPA were injected intraperitoneally (i.p.) every other day. At 22 days after the first i.p. injection of drugs, mice were sacrificed, and tumors were extracted. Each group included five mice.

### 4.11. Statistical Analysis

Results are reported as means of at least three independent experiments (mean ± SD). Data were analyzed using Instat software (Prism5; GraphPad, San Diego, CA, USA). The Student’s *t*-test or one-way analysis of variance with post hoc analysis using Tukey’s multiple comparison test was used for parametric data. Statistical significance was defined as *p* < 0.05.

## Figures and Tables

**Figure 1 ijms-21-02649-f001:**
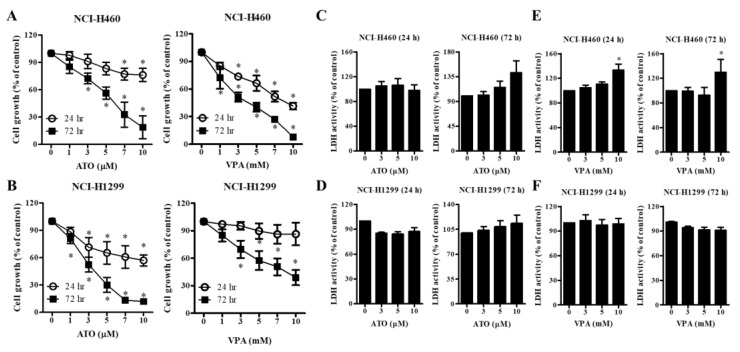
Effects of arsenic trioxide (ATO) and valproic acid (VPA) on cell growth and necrotic cell death in NCI-H460 and NCI-H1299 cells. Exponentially growing cells were treated with the indicated doses of ATO or VPA for the indicated times. (**A**,**B**): Cellular growth changes in NCI-H460 and NCI-H1299 cells as assessed by MTT assays. (**C**–**F**): Necrotic cell death changes as assessed by LDH release and its activity. Lactate dehydrogenase (LDH) activity changes in ATO-treated NCI-H460 cells (**C**) and NCI-H1299 cells (**D**). (**E**,**F**): LDH activity changes in VPA-treated NCI-H460 cells (**E**) and NCI-H1299 cells (**F**). * *p* < 0.05 as compared with the control group. (*n* = 3).

**Figure 2 ijms-21-02649-f002:**
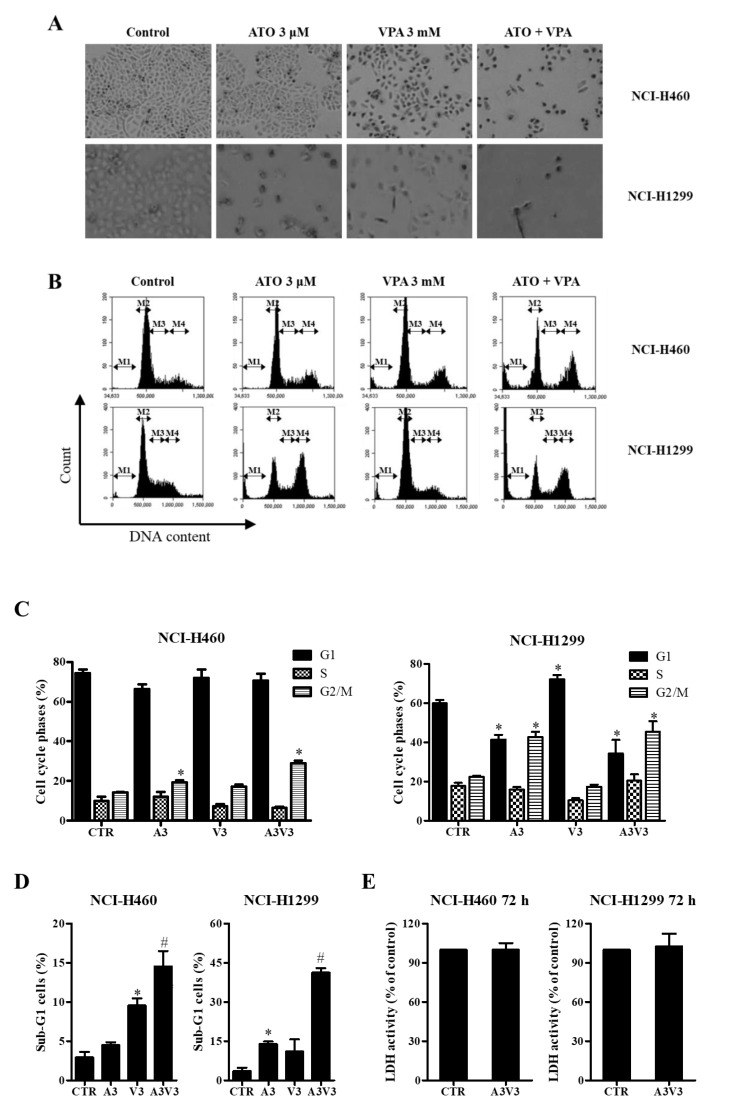
Effects of ATO and VPA alone and in combination on cell morphology and cell cycle distributions. Exponentially growing NCI-H460 and NCI-H1299 cells were treated with 3 µM ATO and/or 3 mM VPA for 72 h. (**A**): Cell morphology changes were captured by an inverted microscope (40×). (**B**): Cell cycle distributions were measured by BD Accuri C6 flow cytometry (M1 regions show sub-G1 cells, M2: G1 phase, M3: S phase, M4: G2/M phase). (**C**): Percentages of G1, S, and G2/M phases in M2, M3, and M4 regions of Figure 2B. (**D**): Percentages of sub-G1 cells in M1 regions of Figure 2B. (**E**): LDH release in NCI-H460 and NCI-H1299 cells co-treated with ATO/VPA. * *p* < 0.05 as compared with the control group. # *p* < 0.05 as compared with cells treated with ATO or VPA. (*n* = 3).

**Figure 3 ijms-21-02649-f003:**
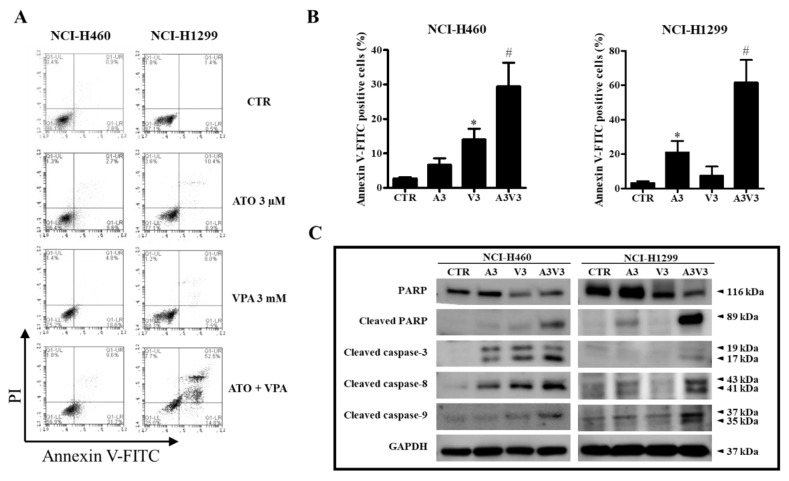
Effects of ATO and VPA alone and in combination on apoptosis and apoptosis-related proteins. (**A**): Exponentially growing NCI-H460 and NCI-H1299 cells were treated with 3 µM ATO and 3 mM VPA for 72 h. Annexin V-FITC/PI staining cells were measured by BD Accuri C6 flow cytometry. (**B**): Percentages of annexin V-FITC positive cells from Figure 3A. (**C**): Expression levels of apoptosis-related proteins as analyzed by Western blot. NCI-H460 cells were treated with ATO and VPA alone and in combination, and NCI-H1299 cells were treated with ATO and VPA alone and in combination. * *p* < 0.05 as compared with the control group. # *p* < 0.05 as compared with cells treated with ATO or VPA. (*n* = 3).

**Figure 4 ijms-21-02649-f004:**
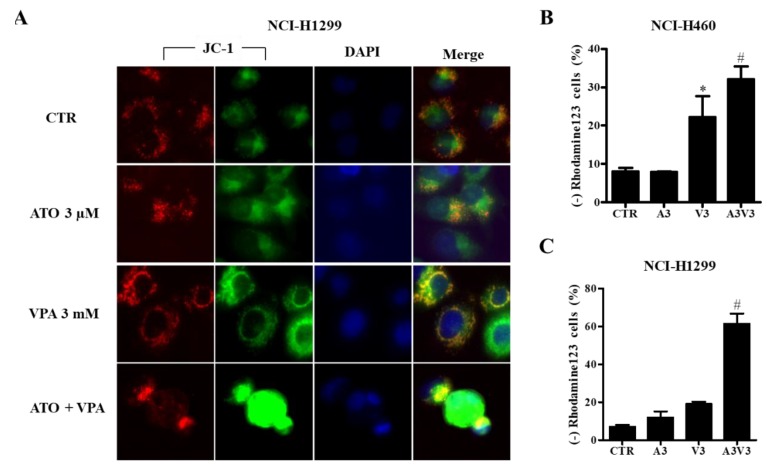
Effect of ATO and VPA alone and in combination on mitochondrial membrane potential (MMP) (∆Ψm). Exponentially growing cells were treated with 3 µM ATO and/or 3 mM VPA for 72 h. (**A**): Representative images of JC-1 (red and green) and DAPI (blue) in NCI-H1299 cells (100×). Red fluorescent images indicate higher MMP (∆Ψm) levels. Green fluorescent images show lower MMP (∆Ψm) levels. (**B**)and **C**: Graphs show the proportions of Rhodamine 123-negative (MMP (∆Ψm) loss) cells in NCI-H460 cells (**B**) and NCI-H1299 cells (**C**) as measured by BD Accuri C6 flow cytometry. * *p* < 0.05 as compared with the control group. # *p* < 0.05 as compared with cells treated with ATO or VPA. (*n* = 3).

**Figure 5 ijms-21-02649-f005:**
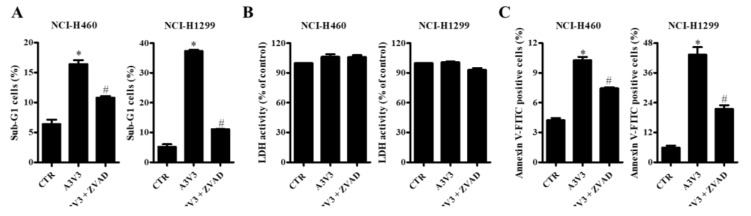
Effect of Z-VAD on cell death in ATO/VPA-treated lung cancer cells. Exponentially growing NCI-H460 and NCI-H1299 cells were co-treated with ATO/VPA in the presence or absence of 15 μM Z-VAD for 24 h. Sub-G1 and annexin V-stained cells were measured with BD Accuri C6 flow cytometry. (**A**): Percentages of sub-G1 cells. (**B**): LDH activity changes. (**C**): Percentages of annexin V-FITC positive cells. * *p* < 0.05 as compared with the control group. # *p* < 0.05 as compared with cells treated with ATO/VPA. (*n* = 3).

**Figure 6 ijms-21-02649-f006:**
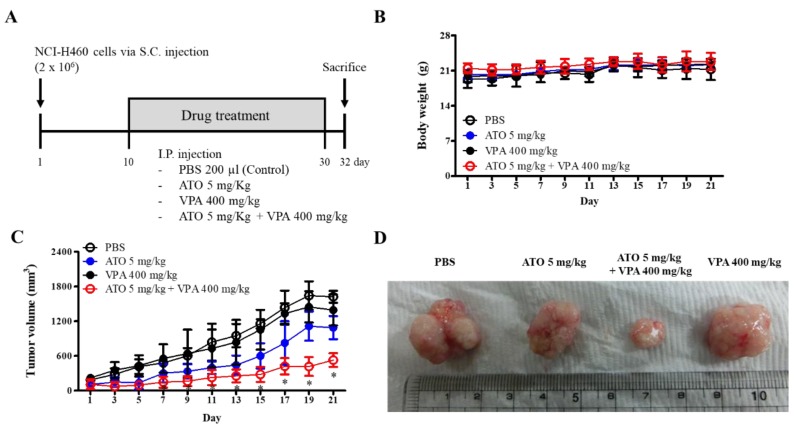
The combination of ATO/VPA inhibits tumor growth in NCI-H460 xenografts from nude mice. (**A**): NCI-H460 cells (2 × 10^6^) were injected s.c. into flanks of 4 week-old female nude mice to establish tumors. Phosphate-buffered saline (PBS) (control), 5 mg/kg ATO, 400 mg/kg VPA, and ATO + VPA were injected i.p. every other day starting on day 10. (**B**,**C**): Body weight changes (**B**) and tumor volume changes (**C**) during 21 days after the first injection of drugs. (**D**): Images of representative tumors of PBS, ATO, ATO/VPA, and VPA injected mice.

**Figure 7 ijms-21-02649-f007:**
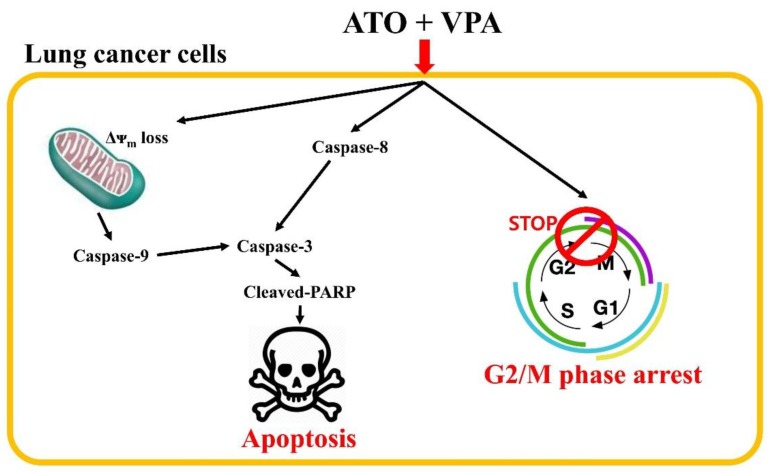
Schematic diagram of ATO/VPA-induced cell growth inhibition in lung cancer cells.

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
