# Peer review of "Combination of Arsenic Trioxide and Valproic Acid Efficiently Inhibits Growth of Lung Cancer Cells via G2/M-Phase Arrest and Apoptotic Cell Death"

_ijms, 2020, doi:10.3390/ijms21072649_

Round 1

Reviewer 1 Report

In the manuscript of Park et al. the synergistic effect of the combination of arsenic trioxide (ATO) and valproic acid (VPA) is investigated on cell growth and apoptosis in vitro and in vivo. The issue is interesting and the methods which were used are well described. Unfortunately, the manuscript has some flaws and the following points should be taken into account:

  • Please clarify and discuss in more detail why lung cancer cells are used as test items, because ATO is mainly associated with leukemia
  • LDH assay is also a cytotoxicity test and it is not useful to detect necrosis. Please explain the missing dose-response effect in the LDH assay
  • Please discuss the used concentration of ATO and VPA. Are these concentrations used in the therapeutically treatment? Which are these concentrations applied in the xenograft model?
  • Please add the n-number of the assays.
  • Please add controls- also positive controls- to proof if the assay works itself
  • Please choose another picture of the western blot of caspase 8 and 9 (NCI-H1299) because the signal is not clear
  • Please add a marker (kDA) to the western blots
  • Please show the complete membrane of the western blot
  • Please add the significance level in Figure 6C and 6D like mentioned in the main text
  • Please discuss and explain the sentence “Apoptosis caused by ATO/VPA apparently involves both intrinsic and extrinsic pathways” in more detail

Reviewer 2 Report

The authors aimed to investigate the effect and mechanism of combination of  arsenic trioxide and valproic acid (ATO/VPA) in lung cancer cell lines. Additionally, they confirmed results on animal model. The authors concluded that the APO and VPA act synergically and theirs combination induces G2/M-phase arrest of the cell cycle and caspase-dependent apoptosis . Overall the results are interesting. The experiments were designed well. However , the manuscript should be rearranged.

Abstract is a summary of results. Pointing out only Ic 50 values of is insufficient in manuscript with running title: Combined effects of ATO/VPA on lung cancer. The authors should remove this part  from abstract.  Additionally, abstracts should present the  backgrounds of the study, aim of experiments, the methods as well as the conclusions.  

The authors should also rearranged of introduction, because the current state of the research field is not fully described. Especially,  introduction has not enough information about valproic acid anticancer activity and about combination treatment  of cancer cells. The authors do not mention the main conclusions. They  should  add it in the end of the introduction.

LDH assay is cytotoxicity assay for assessment of membrane integrity. Damage of cell membrane is effects of cell death not only named necrosis but also is results other types of cell death e.g. apoptosis.  The authors should consider this and  change the description of LDH method and the interpretation of results.

The experimental results are not precisely described. Descriptions of results  are not include values of tested changes.

The results of combined treatments are not compared with other authors results in the discussion. Only effects of alone treatment are discussed.  The authors should develop this part of manuscript.

Round 2

Reviewer 1 Report

Thank you for revising the manuscript, addressing and editing all points.